# STAT3 Role in T-Cell Memory Formation

**DOI:** 10.3390/ijms23052878

**Published:** 2022-03-07

**Authors:** Yaroslav Kaminskiy, Jan Joseph Melenhorst

**Affiliations:** Department of Pathology and Laboratory Medicine, Center for Cellular Immunotherapies, University of Pennsylvania, Philadelphia, PA 19104, USA; yaroslavkaminskiy2018@u.northwestern.edu

**Keywords:** STAT3, T-cell differentiation, immune response, antitumor immunity, cellular therapy

## Abstract

Along with the clinical success of immuno-oncology drugs and cellular therapies, T-cell biology has attracted considerable attention in the immunology community. Long-term immunity, traditionally analyzed in the context of infection, is increasingly studied in cancer. Many signaling pathways, transcription factors, and metabolic regulators have been shown to participate in the formation of memory T cells. There is increasing evidence that the signal transducer and activator of transcription-3 (STAT3) signaling pathway is crucial for the formation of long-term T-cell immunity capable of efficient recall responses. In this review, we summarize what is currently known about STAT3 role in the context of memory T-cell formation and antitumor immunity.

## 1. STAT3 Signaling Overview

STAT3 was discovered in 1994 and initially shown to have transcription factor activity in response to IL-6 and EGF [1,2]. Subsequently, STAT3 was shown to be activated by various growth factors and cytokines acting through the gp130 subunit, common γ and β chain subunits, homodimeric cytokine receptors, type II receptors, receptor tyrosine kinases, and G-protein-coupled receptors [3]. After receptor activation and oligomerization (Figure 1), Janus kinases (JAK1, JAK2, JAK3, TYK2) bind to the receptor, phosphorylate it, and create binding sites for the SH2 domain of STAT3 [4]. STAT3 binds to phosphor-(p)YXXQ motifs on the receptor and is subsequently phosphorylated at Tyr705 by JAK [5,6]. This modification allows STAT3 to dimerize and initiate transcription regulation [7]. Besides Tyr705 residue, Ser727 can also be phosphorylated, which increases STAT3 transcriptional activity [8,9]. Besides being phosphorylated, STAT3 can also be methylated, acetylated, or exert transcriptional activity in an unphosphorylated form [10,11,12,13,14,15,16]. Future studies are required to understand how relevant these modifications are to T cells.

The primary role of STAT3 is transcriptional regulation, and it preferentially binds GAS elements with the consensus motif TTCN_2-4_GAA [17]. STAT3 target genes are numerous, and genome-wide ChIP-Seq analysis of STAT3 in CD4+ showed about 3000 targets [18]. Moreover, unphosphorylated STAT3 can also act as a transcriptional coregulator with collaboration with other proteins [12,15,19].

STAT3 also has important non-translational activities. It was shown to promote mitochondrial metabolism with an effect on other cellular functions [20,21,22]. During the activation of CD4+ T cells, STAT3 accumulates at mitochondria and sustains high mitochondrial membrane potential skewing differentiation toward the Th17 phenotype [23].

STAT3 is pleiotropic and was shown to play a role in physiology and development [24,25]. Its knockout leads to embryonic lethality [26], and STAT3 signaling is essential for cardiac function and maintaining the differentiation potential of stem cells [27,28]. In the blood compartment, STAT3 is required for the proper functioning of both B cells [29,30,31,32] and myeloid cells [33,34,35]. More importantly for the T-cell context, STAT3 signaling is required for hematopoiesis [36,37,38] and the proper development of the thymus [39,40,41] and can lead to abnormalities if perturbed. Moreover, there is increasing evidence that STAT3 signaling plays a role at the late stage of T-cell development [42,43]. For that reason, mouse models with a conditional knockout in immature T cells should be interpreted with caution as the observed phenotype in mature T cells can be affected by prior development abnormalities.

## 2. STAT3 Role in Th17 and Tfh Differentiation

STAT3 is crucial for the differentiation of specific CD4+ T cells subsets. Th17 is a proinflammatory CD4+ effector subset characterized by IL-17 secretion and its differentiation from naïve CD4+ T cells, which occurs in the presence of TGF-β and IL-6 [44,45]. Through STAT3, IL-6 induces the expression of lineage-defining transcription factor RORγt, and IL-21 and IL-23 receptors. IL-21 and IL-23 then create a positive feedback loop and further increase RORγt. STAT3 also induces IL-17 expression in collaboration with RORγt, and STAT3-targeted deletion impairs Th17 differentiation [46,47]. In the presence of IL-6 and/or IL-21 and the absence of TGF-β during priming, CD4+ T cells can differentiate into Tfh (follicular helper) subset [48]. This T-cell subset is characterized by CXCR5 expression and is necessary for B-cell activation and germinal center formation [49]. BCL-6 is the lineage-defining factor of Tfh, whose expression is also positively regulated by STAT3 [48,50]. It is important to note that, in these settings, lineage-defining factors such as STAT3 promote differentiation into specific CD4+ subtypes, in addition to restricting differentiation into alternative subtypes [51,52]. In support of this idea, Martinez-Fabregas et al. (2019) performed STAT3 ChIP-seq and RNA-seq in human Th1 cells and found that, out of 3400 STAT3 bound genes, only 23 were actually upregulated [53]. Th1 differentiation is driven by the STAT4–T-bet signaling axis and suppresses alternative CD4+ T-cell fates [54].

## 3. STAT3 Role in T-Cell Memory Formation

The crucial role of STAT3 in T-cell survival was initially demonstrated by the conditional deletion of this transcription factor in T-cell lineage [55]. In this system, T cells from STAT3-deficient mice were compared with T cells from wild-type mice in vitro. IL-6/STAT3 signaling was shown to have an antiapoptotic effect on T cells independent of Bcl-2 protein. This effect was abrogated in STAT3-deficient cells. IL-6/STAT3 signaling did not have any impact on T cell proliferation. Subsequently, the positive effect of STAT3 signaling on T-cell survival and its lack of proliferation was confirmed in another model [56]. They vaccinated mice with pigeon cytochrome C with or without concurrent IL-6 administration. IL-6 administration did not lead to more significant primary T-cell division, but instead reduced the apoptosis of antigen-specific T cells and increased their numbers 72 days after immunization. Although recall responses of memory CD4+ from both IL-6 treated and nontreated groups were similar, this study shows the STAT3 role in T-cell memory formation in vivo. The survival benefit of IL-6/STAT3 signaling in CD8+ T cells was demonstrated in the adoptive transfer mouse model [57]. After adoptive transfer, IL6Ra+ CD8+ T cells survived longer and had better recall responses than IL6Ra- counterparts. Durant et al. (2010) performed a ChIP-seq analysis of STAT3 in CD4+ and showed that STAT3 binds to Bcl-2 and activates its expression [18]. Moreover, the addition of STAT5-activating IL-2 could not rescue the viability of STAT3^-/-^ suggesting nonoverlapping roles of STAT3 and STAT5 in CD4+ T-cell survival. Another group used conditional knockout of IL-6 receptor alpha chain in double-positive CD4+ CD8+ T-cell precursors and demonstrated that IL-6/STAT3 signaling is essential for proper CD4+ T-cell memory formation [58]. Wild-type or IL6ra^KO^ mice were immunized with OVA peptide with a second immunization 60 days later. Seven days after the second immunization, CD4+ T cells from IL6ra^T-KO^ hosts were analyzed and showed impaired proliferation and cytokine production in vitro. Interestingly, IL6ra and gp130 are downregulated upon TCR engagement in both CD8+ and CD4+ T cells, demonstrating the importance of IL-6 signaling timing [59]. Supporting previous findings, Oh et al. (2011) showed that STAT3 constrains T-cell proliferation but enhances survival. They used CD4-Cre STAT3 conditional deletion to demonstrate STAT3-driven expression of Bcl-2, FoxO1, and FoxO3a. STAT3 directly bound FoxO1 and FoxO3a promoters and STAT3 deficiency caused downregulation of FoxO target genes: p27Kip1 and IkB. The former (p27Kip1) is a cell-cycle inhibitor, and the latter (IkB) is inhibitor of NF-kB, the pathway that is required for the production of mitogenic IL-2.

The role of STAT3 in driving expression of T-cell memory-related BCL6 and SOCS3 was revealed in the acute LCMV mouse model with conditional knockout of STAT3 in CD8+ T cells driven by granzyme B promoter [60]. IL-21 and IL-10 activate STAT3, and the IL-21/IL-10/STAT3/BCL6/SOCS3 axis was shown to be essential for the formation of memory precursors CD8+ T cells (KLRG1- IL7R+). Removal of any component from this axis led to an impaired accumulation of memory precursors T cells. As a result, significantly fewer memory CD8+ T cells formed in the long term. Mechanistically, SOCS3 reduces IL-12-driven phosphorylation of STAT4, which is known to promote short-lived effector (KLRG1+ IL7R-) and suppress memory precursor differentiation [61]. At least in CD4+ T cells, the SOCS3 locus is directly activated by STAT3 binding [18]. In another study, this group immunized mice with dendritic cells in the presence of different adjuvants [62]. They showed that LPS as adjuvant increased serum levels of IL-6 and IL-10 and resulted in more T-cell memory precursors and better secondary immune response. CpG adjuvant, on the other hand, did not induce the expression of either cytokine and, instead, led to inferior memory T-cell formation. The role of IL-6/STAT3 in memory T-cell formation was also indirectly demonstrated when antigen-specific mouse T cells were cultured with APC in the presence of the cytokine [63]. In this system, IL-6 induced expression of ID3, a transcription factor that is essential for proper differentiation of memory T cells. Interestingly, IL-21, which also activates STAT3 and drives memory formation, did not induce ID3 expression. If anything, IL-21 upregulated expression of ID3-antagonizing transcription factor ID2, likely through STAT4 activation. The role of IL-10 in T-cell memory formation was demonstrated earlier in the Listeria infection model [64]. IL-10 deficient mice had a better primary response but fewer memory T cells and suboptimal secondary response.

It is essential to realize that T-cell memory formation negatively correlates with effector function (Figure 2). Key transcription factors (that also initiate epigenetic reprogramming) drive memory differentiation simultaneously blocking terminal effector differentiation [63,65,66,67,68,69]. Depending on the presence, timing, and strength of signals (antigen, costimulatory/inhibitory molecules, cytokines or other signaling molecules, small molecules, ions) that initiate transcriptional reprogramming, T cells can acquire some or all attributes of memory or effector cell lineage [70,71,72,73,74,75,76]. STAT3 signaling is part of the memory formation program and was shown to downregulate T-cell cytotoxic effector functions. Ciucci et al. (2017) cultured CD8+ under Th17 polarizing conditions (IL-6, TGF-b, antibodies against IFNγ, IL-4, and IL-12) and showed that T cells expressing wild-type STAT3 repressed cytotoxic gene expression (GZMB, Pfr1, IFNγ, Tbx21, Eomes), whereas STAT3^-/-^ failed to do so. This effect was mediated, in part, through the STAT3 target gene RORγt [77]. Besides IL-21–IL-10–STAT3–BCL6–SOCS3 axis, there is another mechanistic link between STAT3 and restriction of terminal effector differentiation. STAT3 upregulates expression of FoxO1, which, in turn, represses expression of T-bet, a master regulator of terminal differentiation [78,79].

On the other hand, IL-2/STAT5 signaling is one of the pathways that drive T-cell terminal differentiation and suppress memory formation [68,80,81]. There is evidence that STAT5 and STAT3 act as mutual antagonists. Kalie et al. (2010) demonstrated that on day 3.5 of acute LCMV infection primed T cells can be divided into CD25low and CD25high, which preferentially differentiate into memory precursors (KLRG1- IL7R+) and short-lived effector cells (KLRG1+ IL7R-), respectively [82]. CD25 is the IL-2 receptor alpha-chain and is required for IL-2 signal transduction [83]. CD25low cells had decreased IL-2/STAT5 signaling but were enriched with the IL-6 signaling pathway that acts through STAT3. In another study, the authors compared murine CD8+ T cell ex vivo expansion in the presence of IL-2, IL-15, or IL-21 [84]. Unlike IL-2 and IL-15, which utilize STAT5 for downstream signaling, IL-21 (STAT3-activating cytokine) increased expression of memory-related genes (Lef1, Sell, Tcf7) and repressed expression of effector ones (Eomes, GZMB, IL2Ra). Tcf7 is a direct target of STAT3 and is vital for memory formation [18,66]. Moreover, IL-21 reduces the expression of IL-2-driven Eomes when both cytokines are present simultaneously. Compared with IL-2, IL-21-treated T cells showed superior antitumor activity in the B16 melanoma subcutaneous model. In the acute LCMV model, IL-21 deficiency led to lower numbers of IL-2 producing CD8+ T cells [85]. IL-2 production is a feature of naïve and memory T cells and is repressed in effector cells in a BLIMP1-dependent manner [86]. Interestingly, IL-21 can also induce BLIMP1 expression through STAT3 activation [87]. The authors suggested that an IL-21/BCL6/BLIMP1 incoherent feed-forward loop can explain this discrepancy. IL-21 drives the expression of both BCL6 and BLIMP1 but BCL6 represses BLIMP1 and tunes the balance between these two transcriptional factors.

These studies, however, did not address the molecular mechanism of STAT3–STAT5 antagonism. In the Th17 differentiation setting, this mechanism was studied in greater detail. STAT5 was shown to restrict STAT3-driven RORγt expression and to compete with STAT3 DNA binding at IL-17 locus in CD4+ T cells [88,89]. STAT5 recruitment to the IL-17 locus was associated with decreased H3K4 acetylation and increased binding of NCoR2 corepressor. Although in the Th17 differentiation setting, these studies demonstrate that STAT5 can directly affect STAT3-driven gene expression, one can speculate that STAT5 may have the same direct effect on STAT3-driven genes necessary for memory formation. This idea is supported by a study indicating that IL-2 signaling suppressed BCL6 expression in Th1 cells [90]. The authors showed that STAT5 activated by IL-2 directly bound and repressed BCL6 expression. STAT3 was also shown to share almost 60% of binding sites with STAT5 in murine CD4+ cells [91].

Although the above evidence clearly argues for mutual antagonism, STAT3 and STAT5 can act cooperatively. For example, at least at low doses of IL-2, optimal CD25 (IL2Ra) expression on activated T cells requires intact STAT3 [92]. IL-21-activated STAT3 can positively regulate IL2Ra expression through super-enhancer binding [91]. This is unexpected, given the opposite roles of IL-2 and STAT3 signaling in memory formation.

## 4. Role of STAT3 Signaling in T-Cell Antitumor Immunity

Although the role of STAT3 in memory formation seems to be prominent, its translation into antitumor settings is less clear. There is no consensus on the role of STAT3 in antitumor immunity. Depending on the model, STAT3 activation was either beneficial or detrimental to the ability to fight the tumor. By reviewing studies on STAT3 in cancer settings, we attempt to relate the STAT3 role in memory formation to antitumor immunity. In an early study of STAT3, the Mx1–Cre–loxP system was used to knockout STAT3 in hematopoietic cells in adult mice [93]. STAT3^-/-^ hosts rejected melanoma (B16), and bladder carcinoma (MB49), and these rejections were dependent on the T cells’ presence. STAT3^-/-^ hosts had increased IFNγ production by CD8+ T cells, tumor infiltration, and decreased number of Tregs. This group demonstrated a similar antitumor effect using STAT3 conditional deletion in the T-cell lineage (CD4-Cre) and adoptive transfer model [94]. They also observed increased proliferation of STAT3^-/-^ T cells. In a subsequent study, the same group elucidated the mechanism of STAT3^-/-^ increased antitumor immunity [95]. Using the same STAT3 conditional deletion in the T-cell lineage, they demonstrated that STAT3^-/-^ CD8+ T cells had increased IFNγ production and increased CXCR3 expression in subcutaneous melanoma (B16) and lung carcinoma (3LL) models. IFNγ promoted CXCL10 secretion by myeloid cells in the tumor and, as a result, attracted more CXCR3 expressing effector T cells. The negative effect of STAT3 signaling on CXCR3 expression was confirmed by another study [96]. The authors showed that increased IL17A in serum from colorectal cancer patients activated STAT3, leading to the downregulation of CXCR3 in CD8+ T cells and reduced tumor infiltration. High IL17A levels also correlated with worse patients’ survival. Similar to Yue et al. (2015), induction of cytokine secretion was observed in the glioma model after T-cell transfection with miR-124 [97]. MiR-124 downregulated STAT3 in T cells with a concomitant increase in IL-2, IFNγ, and TNF-α production. However, in another study, miR-124 had an opposite effect on STAT3 signaling [98]. The authors showed that the MeCP2–miR-124 axis suppresses SOCS5 expression in CD4+ T cells. SOCS5 inhibits STAT1 and STAT3 activation, thereby preventing Th1 and Th17 differentiation. The apparent contradiction in the miR-124 role can be attributed to its various effects in CD8+ and CD4+ T cells. Alternatively, STAT1 and STAT3 may have different sensitivities with respect to SOCS5.

In a recent study, the connection between tumor progression and STAT3 driven metabolism was discovered [99]. They showed that in a spontaneous breast cancer model, ablating leptin-driven STAT3 signaling in CD8+ T cells enhanced the antitumor effect by reducing fatty acid oxidation (FAO) and promoting glycolysis. Glycolysis is essential for effector T-cell expansion and synthesis of effector molecules [100,101,102]. Interestingly, they also showed that PD1 signaling, known to induce FAO and inhibit glycolysis, requires intact STAT3 [103]. However, it is not yet clear whether such a metabolic shift restricts antitumor immunity. Published studies demonstrate that it can be beneficial [104,105]. In a recent study, subdermal melanoma (CL24) and pancreatic adenocarcinoma (PanCO2) models were treated with leptin-expressing oncolytic vaccinia virus [106]. Leptin–STAT3 signaling increased mitochondrial content, IFNγ, and TNFa production by CD8+ T cells. As a result, only leptin-engineered viruses improved hosts’ survival. Although confounded by potential leptin effects on tumor cells, this study demonstrates the positive role of T-cell STAT3 signaling in antitumor immunity. As illustrated above, STAT3 signaling restricts glycolysis and promotes FAO in T cells. A similar metabolic shift is also observed during memory formation and was also shown to improve antitumor immunity [107,108].

The role of STAT3 in the antitumor activity of the CD4+ compartment is even more difficult to dissect. Besides memory formation, STAT3 signaling promotes differentiation of naïve CD4+ into Th17 and Tfh. These subsets can have both protumor and antitumor activity depending on the model [109,110,111]. For example, in one study, breast cancer patients had reduced IL6-driven STAT3 phosphorylation in CD4+ T cells. This correlated with a worse survival prognosis and was associated with impaired Th17 differentiation suggesting a decisive role of STAT3 activation [112]. In another mouse model study, IL-6/STAT3 signaling impaired CD4+ T-cell differentiation into the Th1 subset and compromised antitumor immunity [112]. Interestingly, Th17 cells were shown to exhibit prolonged longevity when compared with the Th1 subset [113,114]. Th17 cells also preserved differentiation plasticity and gave rise to Th1-like progeny that exerted antitumor activity. This suggests that Th17 cells are more stem-like than other Th subtypes. From these studies, it appears that Th17 differentiation is interconnected with memory formation potential. Another study showed that the gene expression signature of Tfh is close to the signature of CD8 T-cell memory precursors [115]. Unlike short-lived Th1 cells, Tfh cells were able to become memory cells and (similarly to CD8+ memory precursors) expressed higher levels of BCL6, TCF7, and ID3 and lower levels of IL2Ra and ID2. Thus, additional studies are required to elucidate the pleiotropic role of STAT3 in CD4+ T-cell subsets.

It is useful to look at STAT3 signaling in T cells in the context of a broader question about the role of memory T cells in antitumor immunity. Although not indisputable [116], there are reasons to believe that less differentiated T-cell subsets have superior antitumor efficacy when compared with their more differentiated counterparts [117,118,119,120,121,122,123,124]. Indeed, when preinfusion CAR T cells from 41 chronic lymphocytic leukemia patients were analyzed, CAR T cells from complete responders and partial responders were enriched for memory precursor effector cell (MPEC) genes and IL-6–STAT3 pathway [117]. CAR T cells with favorable CD27+PD1- had increased levels of gp130 expression and STAT3 phosphorylation, indicating the active role, as opposed to the mere biomarker role, of this pathway. It is noteworthy that the IL-6 signaling pathway was also enriched in CD25low effector cells in the LCMV model [82]. These cells were more likely to differentiate into memory precursors as opposed to short-lived effector cells. In a leukemia mouse model, CAR constructs incorporating both STAT5 and STAT3 binding motifs were engineered [125]. It was shown to activate both STAT5 and STAT3 and improve antitumor efficacy in leukemia (NALM6) mouse models. Although in this model, the antitumor effect of STAT3 signaling cannot be separated from STAT5, in vitro stimulated CAR T cells without STAT3 motif had decreased production of IL-2 and a lower number of less differentiated CD45RA+CD62L+CCR7+ CAR T cells. In the subcutaneous mesothelioma (M108) model, CAR T cells expanded in Th17-polarizing conditions provided superior tumor control [126]. A beneficial effect of human T-cell activation in the presence of IL-21 was shown also revealed [127]. In this study, CAR T cells targeting CD19+ were expanded in the presence of only IL-2 or in the presence of IL-2 and IL-21. In the latter case, CAR T cells showed better in vitro proliferation and a significant reduction in the leukemia NALM6 burden (NSG model). Consistent with Hinrichs et al. (2008), IL-21 treatment reduced Eomes expression in vitro. However, as opposed to Hinrichs et al. (2008), it also increased granzyme B expression, highlighting the model differences.

The benefit of less differentiated (memory) T cells for antitumor immunity lies partly in their ability to expand upon antigen encounter significantly and partly in their long-term survival advantage. Some of this survival advantage can be attributed to metabolic flexibility that can be crucial in the hostile tumor microenvironment. STAT3 both drives proper memory differentiation at early stages and modulates T-cell metabolism at the tumor site, contributing to antitumor response. Both aspects lead to better T-cell survival and persistence. On the other hand, memory T cells may have restricted cytotoxic and migration capacity, and in the short term, more differentiated T cells can have an antitumor advantage. However, it is also possible that mouse models are too short of capturing all the dynamics of antitumor response driven by memory T cells and observed in patients. In terms of metabolism, for hypoglycemic tumors, FAO induction seems to be beneficial, whereas, in an environment with sufficient nutrients, it will restrict T-cell expansion by inhibiting glycolysis. Hence, in cellular therapy, it may be reasonable to mimic physiological immune response and combine highly differentiated effector cells with less differentiated memory cells. The former will cause initial tumor reduction and prevent overstimulation of coadministered memory cells. The latter will provide antitumor effects after effector cells contract.

## 5. STAT3 Mutations Linked to T-Cell Memory

Rare Mendelian disorders or naturally occurring mutations in the tumor can provide much information about the physiological role of proteins harboring them. Such mutations can be correlated to their manifestation in cellular phenotype within the physiological environment.

Direct evidence of a role for STAT3 in human T-cell memory formation emerged from a study of individuals with the autosomal-dominant hyper-IgE syndrome (AD-HIES), which is caused by dominant-negative STAT3 mutations [128,129]. Individuals with AD-HIES, among other abnormalities, are susceptible to viral and bacterial infections. Importantly, in such individuals, fewer memory T cells (CD45RO+ CD27+) form, and their naïve T cells have impaired expression of BCL6 and SOCS3, which contribute to memory formation [60,130,131,132]. In line with effector–memory antagonism, increased expression of granzyme B in CD8+ TEMRA and EM subsets from AD-HIES individuals was also demonstrated [129].

Although STAT3 dysregulation often occurs in many cancer types, the role of STAT3 in large granular lymphocytic (LGL) leukemia is of particular interest [133,134]. LGL leukemia is a rare chronic lymphoproliferative disorder of mature T (CD4+, CD8+, or gamma/delta) or NK cells [135,136]. It is an indolent disease with two-thirds of symptomatic cases having splenomegaly, anemia, or neutropenia [137,138]. The origin of the disease is unknown, and the current T-cell LGL leukemia model suggests a two-step mechanism [139,140,141]. First, unknown antigen drives oligoclonal T-cell expansion. Then, dysregulation of survival pathways and, potentially persistent antigen, allow specific clones to avoid T-cell contraction and establish clonal dominance. Importantly, STAT3 seems to play a central role in this survival benefit and was shown to be activated in LGL cells, both by mutations and extrinsic factors, and its inhibition led to apoptosis of T-LGL cells in vitro [142,143,144]. STAT3 mutations appear in up to 70% of T-LGL leukemia cases. Mutations in the SH2 domain are most prevalent, with Y640F being the most common substitution. Y640F, D661V, among other mutations, make STAT3 more prone to dimerization, leading to increased transcriptional activity [145]. On the other hand, T-LGL leukemia cases without STAT3 mutations are still associated with increased STAT3 signaling. High amounts of IL-6 were observed in LGL leukemia individuals and provided antiapoptotic benefits through STAT3 activation [146]. Similarly, the downregulation of SOCS3, capable of reducing STAT3 signaling, was also observed in patients. PTPRT is a phosphatase responsible for the dephosphorylation of Tyr705 on STAT3 with subsequent STAT3 deactivation [147]. Consequently, PTPRT V995M mutation in LGL patients was also associated with increased STAT3 activity (Andersson et al. 2013). BCL11B H126R was associated with increased STAT3 signaling, although there is no known connection between BCL11B and STAT3. Interestingly, in STAT5 Y665F-mutated cases, STAT3 signaling was also activated. This STAT5 mutation is an activating one, suggesting positive crosstalk between STAT3 and STAT5, in contrast to the negative one mentioned in the context of Th17 differentiation [148]. LGL patients with STAT3 mutations are at higher risk of developing rheumatoid arthritis [149]. Interestingly, in a mouse model (adoptive transfer of bone marrow that was transduced with STAT3 mutations) activating STAT3 (Y640F or D661V) mutations by themselves could not induce LGL disease [150]. All in all, the main lesson to learn from TGL leukemia is that STAT3 signaling is associated with a significant survival advantage for T cells, even in the absence of other oncogenic mutations. Such survival is similar to cell survival in memory T cells, which can persist for a very long time after antigen encounter.

STAT3 was reported to be constitutively activated in the late-stage cutaneous T-cell lymphoma (CTCL) [151]. By transfecting tumor cells with dominant-negative STAT3, the authors showed that blocking STAT3 signaling leads to apoptosis. Interestingly, IL-21 expression is also associated with disease progression and with increased mortality in CTCL [152]. It is tempting to speculate that, in this setting, IL-21 activates STAT3 leading to survival benefit.

Mutations in STAT3 are also implicated in the development of autoimmune diseases, including type 1 diabetes [153]. Recently, a mouse model heterozygous for STAT3 K392R gain-of-function mutation was generated [154]. The authors demonstrated that STAT3 K392R mutation in CD8+ T cells counteracts exhaustion, increases cytotoxicity, and can accelerate the development of T1D.

## 6. STAT3 Signaling in the Context of IL-6 Blockade

Cytokine release syndrome (CRS) is a common complication after CAR T product infusion and is accompanied by an extensive release of IL-6 [155]. Now it is successfully treated with tocilizumab (anti-IL-6R antibody) [156]. There is no evidence either from CAR T clinical trials (Kymriah, Yescarta, CD22-BBz CAR T) or from other T-cell-engaging therapies (CD123/CD3 DART or CD19/CD3 BiTE antibodies) that IL-6 blockade has a negative impact on T-cell functionality and antitumor response [157,158,159,160,161,162]. The evidence that IL-6 blockade does not affect therapeutic response implies that at least shortly after infusion, IL-6-STAT3 signaling is dispensable for CAR T cells. This discrepancy is likely explained by the fact that at the end of ex vivo expansion CAR T cells differentiate and lose the expression of the gp130 receptor. Supporting this, Fraietta et al. (2018) showed that only the CD27+PD1- population (least differentiated subset) of CAR T cells expresses gp130 and activates STAT3 upon IL-6 administration [117]. Moreover, gp130 expression is largely restricted to naïve and central memory T-cell subsets from peripheral blood and is coexpressed with CCR7 (unpublished data). However, the presence of soluble gp130 and IL-6Ra receptors in patients after CAR-T infusion raises the possibility that T cells without gp130 expression can still respond to IL-6 [163]. Future studies are needed to further elucidate the role of IL-6 blockade on T cells.

## 7. Conclusions

Considering all the above, there is increasing evidence that STAT3 is required for T-cell memory formation. However, this is largely based on studies that used conditional knockouts in mouse T-cell precursors.

There is a need for cleaner models that interrogate STAT3 in mature T cells. They will eliminate potential effects of perturbed STAT3 that manifested at late stages of T-cell development. One option is CD8a–Cre system (Cre expression driven by the core E8I enhancer and Cd8a promoter), which deletes a gene in mature CD8+ but not CD4+ T cells [164]. GZMB–Cre system can also be used (Cui et al. 2011). Alternatively, one can consider CRISPR–Cas9-mediated gene editing in mature T cells, a system that has been validated in both human and mouse T cells [165,166,167].

STAT3 signaling in human mature T cells is still understudied and most of the evidence is still correlative. Although some parallels between human and mouse models can be drawn, no systematic effort of comparing STAT3 signaling in human and mouse T cells has been made.

In terms of antitumor immunity, STAT3 signaling in T cells seems to be highly context-dependent. There is evidence of both positive and negative roles of this signaling axis in T-cell antitumor immunity. Its negative role can be attributed to STAT3 inhibition of CXCR3-driven tumor trafficking. Hence, inhibition of STAT3 signaling in T cells may be desirable for tumors expressing CXCR3 ligands (CXCL9, CXCL10, CXCL11). Nevertheless, for tumors that do not express these chemokines or blood cancers, STAT3 activation in T cells might provide a persistence advantage with no reduction in trafficking. With STAT3 signaling promoting FAO and restricting glycolysis, STAT3 activation might be beneficial for low glycose environments, where alternative metabolic routes are required for T-cell survival and expansion. On the other hand, if glycose is abundant, STAT3 inhibition can increase T-cell expansion, at least in the short term.

## Figures and Tables

**Figure 1 ijms-23-02878-f001:**
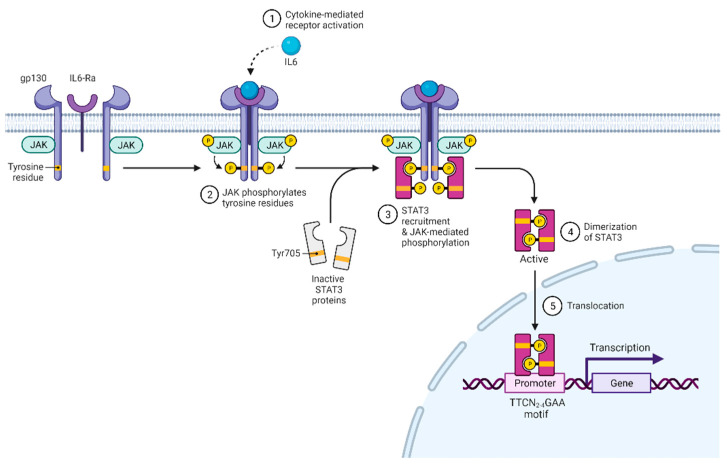
Schematics of cytokine-mediated STAT3 activation. Created with BioRender.com (accessed on 7 September 2021).

**Figure 2 ijms-23-02878-f002:**
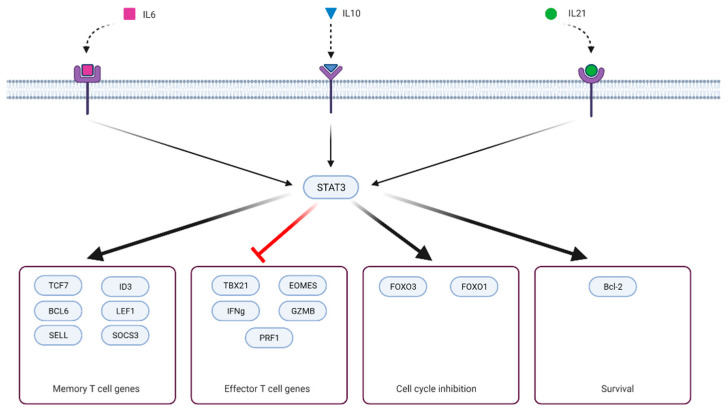
Four groups of genes modulated by STAT3 signaling. Created with BioRender.com (accessed on 7 September 2021).

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
