# Peer review of "STAT3 Role in T-Cell Memory Formation"

_ijms, 2022, doi:10.3390/ijms23052878_

Round 1

Reviewer 1 Report

(1) Overall assessment: In the review article entitled "STAT3 role in T cell memory formation", authors Kaminskiy and Melenhorst provide an in depth review on the role of STAT3 signaling in T cells. The topic is quite niche, but the content of the review is well written and covers the breadth of the topic. It would be of value to those working the field of T cell biology and cytokine signaling, and is worthy of publication.

(2) Criticisms:

        - Given the importance of STAT3 in IL6 signaling, authors should discuss the recent proliferation of IL6 blockade (tocilizumab) in various clinical settings including blocking cytokine release syndrome during CAR-T therapy, and as a therapy for severe COVID-19. In particular, the fact that tocilizumab has not been associated with reduced CAR-T therapeutic efficacy seems to be in disagreement with the discussion within the text wherein  STAT3 signaling is associated with persistent/effective CAR-T responses. Authors should address the effect of tocilizumab on STAT3 signaling and attempt to resolve this apparent conflict.

        - The use of the term "glycose" is unusual, and seems distracting from the content of the text. Authors should consider changing this.

Author Response

Reviewer: "

        - Given the importance of STAT3 in IL6 signaling, authors should discuss the recent proliferation of IL6 blockade (tocilizumab) in various clinical settings including blocking cytokine release syndrome during CAR-T therapy, and as a therapy for severe COVID-19. In particular, the fact that tocilizumab has not been associated with reduced CAR-T therapeutic efficacy seems to be in disagreement with the discussion within the text wherein  STAT3 signaling is associated with persistent/effective CAR-T responses. Authors should address the effect of tocilizumab on STAT3 signaling and attempt to resolve this apparent conflict."

We appreciate the feedback this reviewer provided. We did touch on this important topic of CRS in the following section: 

STAT3 signaling in the context of IL-6 blockade

Cytokine release syndrome (CRS) is a common complication after CAR T product infusion and is accompanied by an extensive release of IL-6 [155]. Now it is successfully treated with tocilizumab (anti-IL-6R antibody) [156]. There is no evidence either from CAR T clinical trials (Kymriah, Yescarta, CD22-BBz CAR-T) or from other T cell engaging therapies (CD123/CD3 DART or CD19/CD3 BiTE antibodies) that IL-6 blockade has a negative impact on T cell functionality and antitumor response [157-162]. The evidence that IL-6 blockade does not affect therapeutic response implies that at least shortly after infusion, IL-6-STAT3 signaling is dispensable for CAR T cells. This discrepancy is likely explained by the fact that at the end of ex vivo expansion CAR T cells differentiate and lose the expression of gp130 receptor. Supporting this, Fraietta et al. (2018) showed that only CD27+PD1- population (least differentiated subset) of CAR T cells expresses gp130 and activates STAT3 upon IL-6 administration [117]. Moreover, gp130 expression is largely restricted to naïve and central memory T cell subsets from peripheral blood and is coexpressed with CCR7 (unpublished data). However, the presence of soluble gp130 and IL-6Ra receptors in patients after CAR T infusion raises the possibility that T cells without gp130 expression can still respond to IL-6 [163]. Future studies are needed to further elucidate the role of IL-6 blockade on T cells.

  • The use of the term "glycose" is unusual, and seems distracting from the content of the text. Authors should consider changing this.

Thank you for pointing this out. We have changed this term in the text.

Reviewer 2 Report

In the manuscript “STAT3 role in T cell memory formation”, the authors proposed to summarize what is currently known concerning the role of STAT3 in the context of memory T cell formation and antitumor immunity. The topic of the manuscript is interesting and the revised version has improved. However, there is still one minor comment to correct.

Specific comments:

  1. Figures have low quality. Please increase the quality of the figures. In addition, add a descriptive legend to the figures, not only titles.

Author Response

IJMS-1452216 mns review

Reviewer 1

Comments and Suggestions for Authors

(1) Overall assessment: In the review article entitled "STAT3 role in T cell memory formation", authors Kaminskiy and Melenhorst provide an in depth review on the role of STAT3 signaling in T cells. The topic is quite niche, but the content of the review is well written and covers the breadth of the topic. It would be of value to those working the field of T cell biology and cytokine signaling, and is worthy of publication.

(2) Criticisms:

        - Given the importance of STAT3 in IL6 signaling, authors should discuss the recent proliferation of IL6 blockade (tocilizumab) in various clinical settings including blocking cytokine release syndrome during CAR-T therapy, and as a therapy for severe COVID-19. In particular, the fact that tocilizumab has not been associated with reduced CAR-T therapeutic efficacy seems to be in disagreement with the discussion within the text wherein STAT3 signaling is associated with persistent/effective CAR-T responses. Authors should address the effect of tocilizumab on STAT3 signaling and attempt to resolve this apparent conflict.

We thank the reviewer for this insightful comment and have added information from various studies addressing the impact of toci-mediated IL6 receptor blockade:

STAT3 signaling in the context of IL-6 blockade

Cytokine release syndrome (CRS) is a common complication after CAR T product infusion and is accompanied by an extensive release of IL-6 [155]. Now it is successfully treated with tocilizumab (anti-IL-6R antibody) [156]. There is no evidence either from CAR T clinical trials (Kymriah, Yescarta, CD22-BBz CAR-T) or from other T cell engaging therapies (CD123/CD3 DART or CD19/CD3 BiTE antibodies) that IL-6 blockade has a negative impact on T cell functionality and antitumor response [157-162]. The evidence that IL-6 blockade does not affect therapeutic response implies that, at least shortly after infusion, IL-6-STAT3 signaling is dispensable for CAR T cells. This discrepancy is likely explained by the fact that at the end of ex vivo expansion CAR T cells differentiate and lose the expression of gp130 receptor. Supporting this, Fraietta et al. (2018) showed that only CD27+PD1- population (least differentiated subset) of CAR T cells expresses gp130 and activates STAT3 upon IL-6 administration [117]. Moreover, gp130 expression is largely restricted to naïve and central memory T cell subsets from peripheral blood and is coexpressed with CCR7 (unpublished data). However, the presence of soluble gp130 and IL-6Ra receptors in patients’ after CAR T infusion raises the possibility that T cells without gp130 expression can still respond to IL-6 [163]. Future studies are needed to further elucidate the role of IL-6 blockade on T cells.

        - The use of the term "glycose" is unusual, and seems distracting from the content of the text. Authors should consider changing this.

We thank the reviewer for carefully scrutinizing our text and have corrected this error.

Reviewer 2

Comments and Suggestions for Authors

This is a reasonably complete review of a relevant question.

We thank this reviewer for this positive feedback.
